# Integration of Selection Signatures and Protein Interactions Reveals *NR6A1*, *PAPPA2*, and *PIK3C2B* as the Promising Candidate Genes Underlying the Characteristics of Licha Black Pig

**DOI:** 10.3390/biology12040500

**Published:** 2023-03-25

**Authors:** Qinqin Xie, Zhenyang Zhang, Zitao Chen, Jiabao Sun, Meng Li, Qishan Wang, Yuchun Pan

**Affiliations:** 1College of Animal Sciences, Zhejiang University, Hangzhou 310058, China; qinqin.xie@zju.edu.cn (Q.X.); 12117017@zju.edu.cn (Z.Z.); barnettca@outlook.com (Z.C.); sunjiabao@zju.edu.cn (J.S.); 2Hainan Institute of Zhejiang University, Building 11, Yongyou Industrial Park, Yazhou Bay Science and Technology City, Yazhou District, Sanya 572025, China; 3Jinan Laiwu Pig Industry Technology Research Institute Co., Ltd., Laiwu 271100, China; 13376342285@163.com; 4Key Laboratory of Livestock and Poultry Resources Evaluation and Utilization, Ministry of Agriculture and Rural Affairs, Hangzhou 310058, China; 5Hainan Yazhou Bay Seed Laboratory, Yongyou Industrial Park, Yazhou Bay Sci-Tech City, Sanya 572025, China

**Keywords:** body length, fat deposition, Chinese indigenous pigs

## Abstract

**Simple Summary:**

Licha black pig is a Chinese breed known for their large body length and appropriate fat deposition. Body length is an important production trait, while fat deposition is known to affect meat quality. However, their genetic characteristics of Licha black pigs have not been well studied. We analyzed the genomic information of 891 individuals of LI pigs, commercial pigs, and other Chinese indigenous pigs, identifying breed characteristics using runs of homozygosity, haplotype, and *F_ST_* selection signatures, which further annotated using protein-protein interaction networks and RNA expression data from FarmGTEx. Our study offers valuable insights into the mechanisms that impact pig body length and fat deposition, which can be applied in breeding to improve meat quality and commercial profitability.

**Abstract:**

Licha black (LI) pig has the specific characteristics of larger body length and appropriate fat deposition among Chinese indigenous pigs. Body length is one of the external traits that affect production performance, and fat deposition influences meat quality. However, the genetic characteristics of LI pigs have not yet been systematically uncovered. Here, the genomic information from 891 individuals of LI pigs, commercial pigs, and other Chinese indigenous pigs was used to analyze the breed characteristics of the LI pig with runs of homozygosity, haplotype, and *F_ST_* selection signatures. The results showed the growth traits-related genes (i.e., *NR6A1* and *PAPPA2*) and the fatness traits-related gene (i.e., *PIK3C2B*) were the promising candidate genes that closely related to the characteristics of LI pigs. In addition, the protein–protein interaction network revealed the potential interactions between the promising candidate genes and the *FASN* gene. The RNA expression data from FarmGTEx indicated that the RNA expression levels of *NR6A1*, *PAPPA2*, *PIK3C2B*, and *FASN* were highly correlated in the ileum. This study provides valuable molecular insights into the mechanisms that affect pig body length and fat deposition, which can be used in the further breeding process to improve meat quality and commercial profitability.

## 1. Introduction

Licha black (LI) pig is an excellent local pig breed in Shandong Province, with close relationships to other Chinese indigenous pig breeds, especially those in Shandong Province. It is reported that LI pig has been domesticated for over 2000 years [1]. However, with the introduction of commercial pig breeds such as Landrace and Yorkshire into China in the 1940s and 1970s, the LI population has shrunk [2]. The number of vertebrae affects their carcass length and meat productivity in pigs [3]. LI pigs have been proven to have lower fat deposition and larger body lengths compared with other indigenous Chinese pig breeds. Compared with most indigenous pigs in China, LI pigs have one or two more vertebrae, which is a specific characteristic in pigs. Yang et al. [2] provided some shreds of evidence that the introgression of the *NR6A1* gene from commercial pig breeds would be related to the increase in the number of vertebrae in LI pigs.

The inbreeding or the introgression of commercial pig breeds into LI pigs may lead to changes in the adaptive characteristics of this breed. Therefore, it is of great significance to explore specific genes for genetic conservation and improvement of LI pigs. Previous studies reported that the effective population size (Ne) of the LI population is estimated to be 8.7, which is lower than many local pig breeds in China; meanwhile, the genomic inbreeding coefficient based on ROHs is estimated to be 0.11 [4]. 

A set of candidate genes are significantly associated with body length and fat deposition traits [5,6,7]. For example, *NR6A1* is the candidate gene for the rib trait, and *VRTN* is the candidate gene for the thoracic vertebra trait [5,6]. Furthermore, *MSRB2* and *PLAG1* are strong candidate genes for back thickness [7]. However, there is a lack of systematically investigating of the genetic mechanisms underlying body length and fat deposition traits of LI pigs. Therefore, it is necessary to use a variety of advanced research methods to study the germplasm characteristics and gene operation mechanism of LI pigs to uncover the specific characteristics of the breed.

In this study, we aimed to detect and evaluate the germplasm specificity of the LI pigs and investigate candidate genes closely associated with their genetic characteristics, particularly body length and fat deposition, using various strategies, including runs of homozygosity, haplotype, and selection signals. We also aimed to compare and predict the specific genetic basis of LI pigs using the distribution pattern of alleles and gene expression. To enhance the credibility of our conclusion, we combined the prediction of amino acid transformation of variants, protein–protein interaction network, and transcription level analysis.

## 2. Materials and Methods

### 2.1. Populations and Data

We utilized data from the “Zhongxin-I” Porcine 50 K SNP Chip (Beijing Compass Agritechnology Co., Ltd., Beijing, China) and whole genome re-sequencing (WGS) of 891 pigs. The sample set included 220 LI pigs (11 were newly generated WGS data, and 209 were downloaded 50 K SNP chip data [4]), 305 commercial pigs (23 were newly generated WGS data, and 282 were downloaded WGS data from public databases), i.e., Duroc (D), Landrace (L), Yorkshire (Y), and 366 Chinese indigenous pigs except for LI (197 were newly generated WGS data, and 169 were downloaded WGS data from public databases) from 11 provinces, i.e., Anqing Six-end-white (AQ), Bamaxiang (MX), Bamei (BM), Dahe (DH), Diannan Small-ear (DN), Erhualian (EH), Jiangxing Black (JX), Jinhua (JH), Laiwu (LW), Meishan (MS), Ningxiang (NX), Tongcheng (TC), Wannan Black (WN), Wuzhishan (WS), Xiang (XZ). Appendix A listed the details of the sampled pig breeds in this study, including breed names, abbreviations, and sample sizes. The data sources and bioproject numbers were mentioned in Data Availability Statement.

We strictly followed the Animal Care and Use Committee of Zhejiang University (Hangzhou, China) under permission No. ZJU20220224 for all experimental procedures related to the newly generated WGS data. We collected ear tissue samples from 231 pigs from the native pig conservation farm for high-throughput re-sequencing. After extracting DNA using the Qiagen DNeasy Tissue kit (Qiagen, Dusseldorf, Germany), we verified the integrity and purity of the DNA using agarose gel electrophoresis and A260/280 ratio. We then performed end-repair, A-tailing, ligation of pair-ended adapter, and size-selection for sequencing and amplification of the genomic DNA with the Covaris system. Finally, we sequenced the amplified fragments on the HiSeqX platform using the scheme recommended by the manufacturer of Novogene (Beijing, China). The processing process refers to PHARP [8]; briefly, we used BWA v0.7.17 [9] software for mapping, SAMtools v1.10 [10] software for sorting, and GATK v4.1.6 [11] software for calling SNP and VariantFiltration. We used the following criteria for VariantFiltration: “QD < 2.0, FS > 60.0, MQ < 40.0, SOR > 3.0, MQRankSum < −12.5, ReadPosRankSum < −8.0”. We detected a total of 53,789,547 SNPs from the newly generated WGS data, including 231 individuals.

We processed the downloaded WGS data of 451 individuals with GATK v4.1.6 [11] and obtained 53,789,547 SNPs, the same as the newly generated WGS data. We consolidated all the WGS data and performed basic filtering using PLINK v1.9 [12] (command: PLINK --geno 0.1 --mind 0.1 --maf 0.05), which retained 28,490,962 SNPs from the WGS data of 11 LI pigs, 366 Chinese pigs, and 305 commercial pigs. We used the filtered WGS data for “*screening of candidate genes using whole genome sequencing data* (in Section 2.6)” (Figure 1).

We used the WGS data as a panel to imputed chip data of 209 LI pigs with Beagle v5.4 [13] software, resulting in 53,789,547 SNPs. We then filtered the imputed data with BCFtools v1.10.2 [10] to remove SNPs with DR^2^ (Dosage R-Squared) less than 0.3 and merged it with the WGS data. To ensure high data quality, we performed quality control with PLINK v1.9 [12] software and eliminated unqualified individuals and SNP sites based on the following criteria: (1) Genotype detection rate and individual detection rate were greater than 95%; (2) The loci with the minimum allele frequency (MAF) less than 0.05 were filtered out; (3) Remove SNPs on sex chromosomes. The combined data retained a total of 277,882 high-confidence SNPs from 891 individuals, which we used for (1) *runs of homozygosity (ROH) and haplotype detection among LI pigs* in Section 2.3, and (2) *identification of functional annotation of selection signatures* in Section 2.4 (Figure 1). We observed that these SNPs were distributed roughly uniformly across the genome, indicating that they can represent the information of the whole genome (Appendix A). 

We further pruned the combined data by discarding SNPs whose LD (linkage imbalance) was greater than 0.4 in these populations (command: PLINK --indep-pair 50 10 0.4), leaving 29,934 SNPs in total. The LD-pruned data were used for “*Population structure and genetic diversity* (in Section 2.2)” (Figure 1).

### 2.2. Population Structure and Genetic Diversity

To illustrate the genetic diversity and population structure between these populations, we conducted the following steps using LD-pruned data (891 individuals, 29,934 SNPs): (1) Genetic distance between populations was calculated by PLINK v1.9 [14], applied to construct NJ-trees (neighbor-joining trees) by MEGA11 [15], and illustrated using iTOL [16]. (2) Principal component analysis (PCA) was often used to detect and quantify the population’s genetic structure in population genetics. PCA was calculated using PLINK v1.9 [14]. (3) The population structure analysis was conducted using ADMIXTURE v1.3.0, and the range of the ancestor cluster (K) was set from 1 to 20. The results are displayed using the R package POPHELPER [17]. (4) TreeMix v1.13 [18] was used to perform historical population migration. 

### 2.3. Runs of Homozygosity and Haplotype Detection among LI Pigs

We utilized the combined data of 220 LI individuals, comprising 277,882 SNPs, for ROH and haplotype. We detected runs of homozygosity for each individual using “-homozyg” of PLINK v1.9 [14], adhering to the criteria established by Zhang et al. [19]: the size of the ROH sliding window was 50 SNPs, each window allows 3 heterozygotes or 1 deletion, the minimum SNP density in the ROH was 1 SNP/100 Kb, and the interval between two consecutive SNPs is less than 250 Kb. We selected the SNPs with the top 1% frequency in all individual ROH regions and merged them into each continuous section, which is called ROH island. All homozygous segments were classified into three categories according to segment length: 1–5 Mb (${ROH}_{1–5\mathrm{Mb}}$), 5–10 Mb (${ROH}_{5–10\mathrm{Mb}}$), and greater than 10 Mb (${ROH}_{>10\mathrm{Mb}}$), respectively.

For haplotype block estimation, we employed PLINK v1.9 [14] with the parameter “--blocks no-pheno-req --blocks-max-kb 200”. The standard was as follows: the upper limit of a block length is 200 Kb. Then, we calculated the length, frequency, and distribution of haplotype blocks in both varieties and selected the top 1% haplotype blocks as the candidate region.

### 2.4. Identification of Functional Annotation of Selection Signatures

We processed the combined data of 891 individuals (220 LI, 305 commercial pigs, and 366 Chinese indigenous pigs), which contains 277,882 SNPs, for *F_ST_* analysis. To determine the highly differentiated genome regions of LI pigs, other Chinese pigs, and commercial pigs, we used vcftools v0.1.13 [20] to calculate *F_ST_* and identify the significant differentiation loci between these pigs. According to the *F_ST_* value of SNP, the first 1% of SNP sites were selected as the final candidate regions. For candidate regions, such as ROH islands, haplotype blocks, and *F_ST_* significant differentiation loci, we used the Sus scrofa reference genome 11.1 of Ensembl Genome Browser to annotate the genomes of candidate regions and combine R v4.2.1 [21] package GALLO [22] to search for genes.

### 2.5. RNA Expression Specificity of Candidate Genes in Human and Pig Tissues

To identify whether the RNA expression of these candidate genes was tissue-specific, we confirmed the RNA expression in pig tissues according to the Pig RNA Atlas (https://www.rnaatlas.org/, accessed on 1 June 2021) [23]. We then collected the candidate genes identified from different candidate regions and visualized the results using a Venn diagram generated with the R v4.2.1 [21] package VennDiagram [24]. Our focus was on the shared genes detected by two or more methods, and we further investigated the function and the RNA expression level in different tissues.

### 2.6. Screening of Candidate Genes Using Whole Genome Sequencing Data

We utilized the filtered WGS data, containing 28,490,962 SNPs from a total of 682 individuals (11 LI pigs, 366 Chinese pigs, and 305 commercial pigs), to identify the target gene. To validate our findings, we followed these steps: (1) Fisher test was used to analyze whether the distribution (type and quantity) of reference base (REF) and alternate base (ALT) corresponding to each locus allele of LI breed and other breeds (other Chinese breeds or commercial breeds) were significantly different. The results with a *p*-value < 0.05 were reported to identify potential variation loci in the LI breed, hereinafter referred to as significant loci. (2) Proportion of significant loci in the whole gene. (3) With Ensembl Variant Effect Predictor (VEP) [25], we evaluated the consequence and its impact on the variants at the significant loci. 

### 2.7. Verification of Gene Influence and Mechanism

We conducted three methods to explore the mechanism of growth and fatness traits-related genes. The methods were as follows: (1) For each variant mapped to the reference genome, we used VEP to recognize amino acid transformations caused by variants. (2) Protein–protein interaction (PPI) network was constructed by GeneMANIA (https://genemania.org/, accessed on 16 January 2023) [26]. We obtained a variety of network categories, including co-expression, physical interaction, genetic interaction, shared protein domains, co-localization, pathway, and enriched genes in the network. (3) The Farm animal Genotype-Tissue Expression TWAS-Server (FarmGTEx TWAS-Server, http://twas.farmgtex.org/, accessed on 12 December 2022) [27] provides the TPM (transcript per million) of numerous genes across pig tissues. The TPM data were used to evaluate the correlation of gene expression between *NR6A1*, *PAPPA2*, *PIK3C2B*, and *FASN* in different pig tissues. The results with a *p*-value < 0.05 were reserved.

## 3. Results

### 3.1. Population Relationship and Structure

We used 29,934 SNPs after LD filtering to analyze the relationship between LI and other populations. To examine the phylogenetic relationships of these 891 pigs, a genetic-distance-based NJ-tree analysis was conducted (Figure 2A). The NJ-tree showed that individuals were subdivided into three main branches representing LI, other Chinese breeds, and commercial breeds. LI pigs could form a branch different from other pigs in China and foreign countries, which showed that LI had unique genetic characteristics. The individuals from the same breed clustered into the same groups, except for the MS, XZ, WS, and JX, which were classified into the same pig strain. To further determine whether LI was a unique Chinese genetic resource, a principal component analysis was performed. PC1, which accounted for 45.33% of the total variance, proved the similarity of the chip and the re-sequencing data in the principal components, separating the LI pigs and commercial pigs (Figure 2B). Except for LI pigs, all individuals from the Chinese pig breeds were clustered together. PC2 and PC3 explained 22.76% and 11.22% of the total variance, respectively. Then, the ADMIXTURE could further observe the lineage composition of LI pigs. When two ancestors were assumed, Chinese indigenous pigs and commercial pigs were clearly distinguished (Appendix A), but LI and some other Chinese pig populations contained some genetic ancestry that was similar to commercial populations, especially for AQ, LWU, and DH. LI was always the most admixed population over different K values. When K was 8, the results presented by the data of two batches of LI pigs were slightly different.

### 3.2. Variety Specificity Reflected in ROH Islands, Haplotype Blocks, and F_ST_ Analyze

In our study, we detected a total of 31,892 ROH fragments from 277,882 high-confidence SNPs in 220 LI pigs, with an average length of 2.009 Mb. All ROH fragments covered 19.314% of the whole genome. Among the three types of ROH fragments, the largest number of ${ROH}_{1–5\mathrm{Mb}}$ is 30,905, accounting for 96.905% of the total number of ROH fragments, and the genome coverage was 15.748% (Appendix A). The distribution of ROH fragments with chromosomes showed that SSC1 had the largest number of ROH fragments, whereas the number of ROH fragments in SSC12 was the smallest (Appendix A). To compare the differences in ROH fragments among varieties, we constructed a dot map of the number and length of ROH from individuals of different breeds (Appendix A). The figure indicated that the number and length of ROH of all individuals in the LI population were at a low level, which might suggest that the inbreeding degree of LI pigs was also at a low level. To identify the genomic regions with high ROH frequency, we calculated the frequency of SNP occurrence in the ROH and selected the top 1% among those with the highest frequency (Figure 3A). Our study detected a total of 35 ROH islands, and there were 42 genes within those regions (Appendix A).

We identified a total of 14,482 haplotype blocks containing 264,823 SNPs in the LI population, which spanned a genomic region of 707,157.385 Kb, as presented in Appendix A. The largest number of haplotype blocks (1716) and the longest coverage area (91,328.017 Kb) were found in SSC1, while SSC17 exhibited the least haplotype blocks (382), and SSC12 had the shortest haplotype block coverage chromosome area (14,102.584 Kb). Additionally, SSC1 contained the largest number of SNPs (32,462) and the highest proportion of SNPs in chromosomes (12.258%). About 147 haplotype blocks with the first 1% of the length were selected as possible candidate regions, and there were 210 genes within those regions, as shown in Table 1 and Appendix A.

We obtained a total of 277,882 autosomal SNPs from 891 pigs. The threshold values of the first 1% selective signal SNPs are defined as 0.723 (between LI pigs and other Chinese pigs) and 0.605 (between LI pigs and commercial pigs), respectively (Figure 3B,C). Unlike ROH and haplotype analyses, which enable us to find common features within the LI breed, detecting *F_ST_* signal between LI pigs and others can identify the differentiated features of LI pigs by comparing them to other Chinese indigenous pigs or commercial pigs. Ultimately, we identified 362 candidate genes and 326 candidate genes from 2779 important SNPs (Appendix A), with the former resulting from the selection signal comparison (F*_ST_*-1) of LI pigs and other Chinese pigs and the latter from the selection signal comparison (F*_ST_*-2) of LI pigs and commercial pigs.

### 3.3. Candidate Genes Related to Growth, Muscle, and Fat

To further reduce the false positive rate detected by different analysis methods, we took the genes detected two or more as candidate genes (obtained from ROH, haplotype, and *F_ST_* analysis) for further research. Finally, 53 genes met the screening requirements. Appendix A was the Venn diagram constructed according to the candidate genes obtained from three methods. The candidate genes detected were correlated with many interesting traits, such as growth (*DGKB*, *IFT81*, *KCNH7*, *KCNIP4*, *PAPPA2*, *PGM5*, and *ZFAT*), muscle (*ANKRD44*, *HDAC11*, *LPP*, *PGM5*, *PIK3C2B*, and *TRIM54*), and fat (*DGKB*, *HDAC11*, *KCNH7*, and *PIK3C2B*). Table 2 shows the functions of twelve important genes related to growth, muscle, and fat and the tissues where these genes had specific RNA expression in pigs and humans. Among the undetected genes, there were also genes related to many important economic traits, such as *NR6A1*, a gene related to the number of vertebrae.

### 3.4. Allele Distribution and Variation of NR6A1, PAPPA2, and PIK3C2B

We further screened the above 13 candidate genes using the filtered WGS data of 11 LI pigs, 305 commercial pigs, and 366 Chinese indigenous pigs (excluding LI pigs). We used the Fisher test to analyze whether there were significant differences in the type and quantity of reference base (REF) and alternate base (ALT) of LI breed and other breeds with the change of the locus. We identified several loci that were significantly (*p*-value < 0.05) different between LI breed and other breeds, which we refer to as significant loci. Table 3 recorded the information on some significant loci of candidate genes, including the type and number of alleles. Then, we found that the percentage of significant loci of *KCNH7* and *NR6A1* in LI breeds was higher than that of other Chinese breeds, while the percentage of significant loci of *PAPPA2* and *PIK3C2B* was higher than that of commercial breeds (Appendix A). Furthermore, to assess the consequence and impact of the variants, we used Ensembl Variant Effect Predictor (VEP) to screen 1, 7, and 3 loci with moderate or higher impact in *NR6A1*, *PAPPA2*, and *PIK3C2B* three genes, respectively (Table 3). *NR6A1* had a missense variant at 265,347,265bp on chromosome 1, and *PAPPA2* and *PIK3C2B* had 7 and 3 missense variants as well. A missense variant was a sequence variant that changed one or more bases, resulting in different amino acid sequences without changing the length.

### 3.5. Amino Acids Transformation and Interaction of NR6A1, PAPPA2, and PIK3C2B

The amino acid transformation of the variants was showed in Table 3. Each variant was capable of converting one or more amino acids. As shown in the table, the variant of *NR6A1* at SSC1: 265347265 caused the transformation between proline (P) and leucine (L). Seven variants of *PAPPA2* were responsible for the conversion of seven amino acids, while *PIK3C2B* only showed two possible amino acid conversions.

Figure 4 shows the candidate gene interaction network (including *NR6A1*, *PAPPA2*, and *PIK3C2B*) constructed through the GeneMANIA database, as well as the possible pathways significantly related to growth, muscle, and fat. In PPI (protein–protein interaction) network, *PAPPA2* (Figure 4B) displayed fat-related and growth-related functions, such as regulation of insulin-like growth factor receptor signaling pathway, regulation of smooth muscle cell migration, and growth factor binding, which might be related to the interaction with *IGF1*, *IGFBP3*, *IGFBP4*, *IGFBP5*, and *IGFALS*. *PIK3C2B* (Figure 4C) drew the interaction network related to the function of the glycerophospholipid biosynthetic process, glycerolipid biosynthetic process, lipid modification, phospholipid biosynthetic process phosphatidylinositol metabolic process, and all functions show interesting possibilities with lipid synthesis and occurrence. Moreover, the possible genes that *PIK3C2B* interacted with have also been excavated, providing strong evidence for further explaining the possibility of *PIK3C2B* in function. Furthermore, the network formed by *NR6A1*, *PAPPA2*, and *PIK3C2B* explained the *FASN* gene associated with the three (Figure 4A). *NR6A1* and *PAPPA2* were co-expressed with *FASN*, while *PIK3C2B* was a physical interaction relationship with *FASN*.

To further verify that the four genes are indeed related, we calculated the correlation coefficient of gene expression TPMs between *NR6A1*, *PAPPA2*, *PIK3C2B*, and *FASN* across pig tissues (Figure 5). The correlation coefficients of gene expression between four genes in twelve tissues were significant (*p*-value < 0.05). The study demonstrates a positive correlation between three genes (*NR6A1*, *PAPPA2*, and *PIK3C2B)* and *FASN* in various pig tissues. Notably, the correlation was high in the ileum (ranging from 0.936 to 0.972) and moderate in the small intestine (ranging from 0.644 to 0.692), cartilage (ranging from 0.670 to 0.771), and lung (ranging from 0.656 to 0.898).

## 4. Discussion

### 4.1. Genetic Characteristics of Licha Black Pigs

Our study revealed that the LI pigs tended to cluster together, as shown by both the NJ-tree and PCA results. Furthermore, the LI pigs formed a distinct group that was closer to other Chinese indigenous breeds and farther away from commercial breeds. The admixture analysis suggested that the LI breed might have been mixed with Duroc and Landrace. Given that Shandong Province is a major agricultural province in China, there is a high likelihood of gene exchange occurring among local pig breeds. Moreover, historical records indicated that commercial breeds were introduced in the early 20th century to improve the breed structure, indicating the LI breed may have a lineage of commercial breeds. Interestingly, our analysis revealed some differences Wang et al. [4] and our WGS data, as demonstrated in the admixture results (Appendix A). These differences should be taken into account when evaluating the characteristics of the LI breed.

### 4.2. Population-Specific Genes in Licha Black Pigs

In our study, we used ROH islands, haplotype blocks, and *F_ST_* to identify population-specific genes of the LI breed, resulting in 738 candidate genes. Among these genes, some could explain the unique characteristics of the LI breed. For example, *DMRT1* was a gene related to sexual regulation. *DMRT1* was essential for maintaining mammalian testicles [52] and maintaining gonadal sex long after birth [53]. To shed light on the body length characteristics of the LI breed and speculate on its impact on muscle and fat, we finally locked thirteen key candidate genes related to growth, muscle, and fat. These genes included nine candidate genes related to growth traits (*DGKB*, *IFT81*, *KCNH7*, *KCNIP4*, *NR6A1*, *PAPPA2*, *PGM5*, and *ZFAT*), six candidate genes related to the muscle (*ANKRD44*, *HDAC11*, *LPP*, *PGM5*, *PIK3C2B*, and *TRIM54*), and four candidate genes related to fat (*DGKB*, *HDAC11*, *KCNH7*, and *PIK3C2B*). 

*NR6A1* (Nuclear Receptor Subfamily 6 Group A Member 1) was a protein-coding gene. It has been demonstrated that *NR6A1* was a strong candidate for body length in domestication pigs [3] and other vertebrates [54,55]. The increase in body length may be reflected in the increase in the number of vertebrae, nipples, and ribs [56]. Consistently, a study on the characteristics of the spine of LI pigs showed that *NR6A1* is a gene related to the vertebrae number, which was associated with individual growth traits [5]. Moreover, this gene seems to be fixed in some commercial, domestic pigs [57]. *PAPPA2* (Pappalysin 2) showed the regulatory ability related to biological growth. A study on *PAPPA2*-deficient humans showed progressive growth failure, accompanied by impaired *IGFBP* protein hydrolysis and a significant increase in the total concentration of *IGF-I* [33]. Previous studies suggested that *PAPPA2* deficiency caused *IGFBP* regulation defects, leading to inefficient *IGF-I* bioavailability and longitudinal growth disorders [34,58]. Nesrin et al. [37] showed that *PIK3C2B* (Phosphatidylinositol-4-Phosphate 3-Kinase Catalytic Subunit Type 2 Beta) could trigger specific muscle ablation and is identified as a genetic modifier of *MTM1* mutation, and here we know that mutation of *MTM1* is the cause of myotubular myopathy. On the other hand, some studies have confirmed that heterozygous ultrarare variants in *PIK3C2B* could cause defective lipid signals [38]. According to these findings, the changes in muscle and fat in vivo may be related to the differential expression of *PIK3C2B*.

### 4.3. NR6A1 Variant Is a Potential Selective Marker for Improving Pig Body Length

In our study, we identified 989 loci of the *NR6A1* gene that were specific to the LI breed compared with commercial breeds and 525 loci that were specific to the LI breed compared with other Chinese breeds. Notably, the locus located at SSC1:265347265 displayed a significant difference in base composition between the LI breed and other breeds. The proportion of the alternate base at this locus in the LI breed was between that of commercial breeds and other Chinese breeds, suggesting that this locus may have been influenced by commercial breeds. Using VEP, we predicted that the variant of *NR6A1* at SSC1:265347265 is a missense variant with moderate impact, leading to the transformation between proline and leucine. This amino acid transformation may make the LI breed more similar to commercial breeds. Based on the previous study [2], we speculated that this variation was probably due to the introgression of the commercial *NR6A1* haplotype into the LI breed. In the process of breed infiltration, commercial pigs could introduce more A bases into the SSC1:265347265 locus of the LI breed, resulting in the change of coding amino acids. Our results indicated that the LI breed’s advantage in body length, which distinguishes it from other Chinese breeds, might be due to the variation of *NR6A1* at locus SSC1:265347265.

### 4.4. Interaction of NR6A1, PAPPA2, and PIK3C2B Affect Pig Fat Deposition

We describe the candidate genes in the PPI network by identifying the genes that have the largest number of significant associations with other genes. As one of the central genes, *PAPPA2* displays growth-related functions like growth factor binding under co-expression and physical interaction with *IGF1*, *IGFBP3*, *IGFBP4*, and *IGFBP5*. Hence, we believe that *PAPPA2* may be an important gene affecting pig growth or body length as well. Another central gene, *PIK3C2B*, is more enriched in the function and pathway of lipid expression. Therefore, we speculate that *PIK3C2B* may be one of the important genes affecting fat deposition or lipid metabolism in pigs. However, understanding the related functions of *NR6A1*, *PAPPA2*, and *PIK3C2B* led us to explore the proteins affected by these three genes. Here, the network formed by *NR6A1*, *PAPPA2*, and *PIK3C2B* explains the gene that is associated with the three, *FASN* (fatty acid synthase), which has been confirmed to be possibly related to intramuscular fat and fatty acid in cattle and sheep [59,60]. Thus, the interaction of *NR6A1*, *PAPPA2*, and *PIK3C2B* may be related to the expression of *FASN*.

The correlation of gene expression of *NR6A1*, *PAPPA2*, *PIK3C2B*, and *FASN* varies with the change of pig tissues. The results show that the RNA expression levels of *NR6A1*, *PAPPA2*, *PIK3C2B*, and *FASN* genes have a strong correlation in the ileum. Fat-related genes are highly expressed in the small intestine, including the duodenum, jejunum, and ileum, and the small intestine is the tissue where fat begins to be digested [61]. Based on our analysis, we speculated that *NR6A1*, *PAPPA2*, *PIK3C2B*, and *FASN* might cooperate in carrying out a biological process in the ileum that was probably related to the digestion and absorption of fat, according to our inference. In this case, the growth (body length) traits of the LI breed may indeed have an unknown association with fat deposition. It is not yet known that this is a linkage dominated or interacted by one party, which may need further experimental verification.

## 5. Conclusions

Our study supports the theory that LI is a Chinese indigenous pig with germplasm characteristics but with some infiltration from commercial breeds. Additionally, we identified specific genes related to growth, fat, and muscle in LI pigs, such as *NR6A1*, *PAPPA2*, and *PIK3C2B*. *NR6A1* has been linked to the growth and body length of pigs. Meanwhile, *PAPPA2* and *PIK3C2B* are newly discovered differential genes in Chinese indigenous pigs that may also affect growth and fat deposition, respectively. We also found that the allele distribution of LI suggested the possibility of *NR6A1* variants, particularly the effect of base A in SSC1:26534726 in the selection process, which may have been introduced through haplotype from commercial pigs. The change of coding amino acid caused by this variant may promote a slow change in pig body length. In conclusion, our study contributes to the research on the genetic characteristics of LI pigs and highlights the potential role of *NR6A1*, *PAPPA2*, and *PIK3C2B* in the longer body length and appropriate fat deposition of the LI.

## Figures and Tables

**Figure 1 biology-12-00500-f001:**
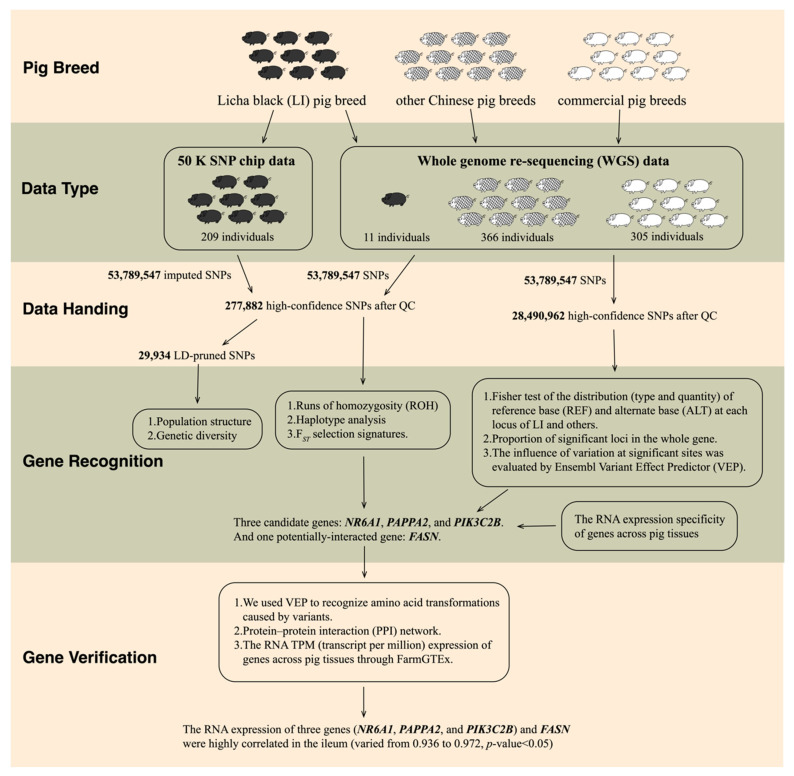
Process and data of this study.

**Figure 2 biology-12-00500-f002:**
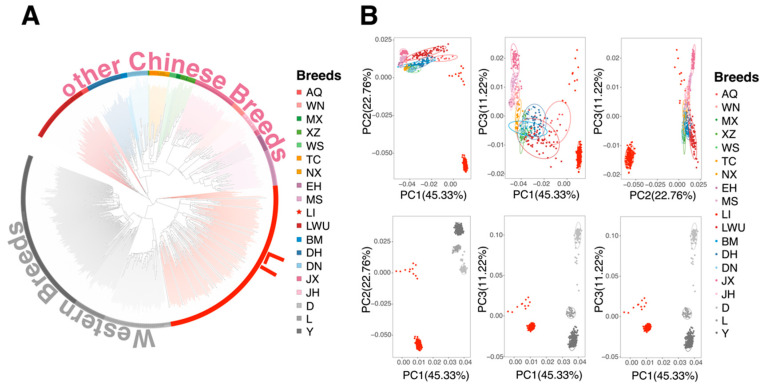
Population relationship and structure. (**A**) NJ-tree based on an identical-by-state matrix among 891 pigs tested in this study. Each abbreviation represents a pig breed, and each color represents an ecotype (Appendix A). (**B**) Principal component analysis of Licha black (LI) pig and other Chinese breeds (up) and PCA of Licha black pig and foreign Chinese breeds (down). Each figured point represents the eigenvector of one individual. The first (PC1), second components (PC2), and third components (PC3) are shown.

**Figure 3 biology-12-00500-f003:**
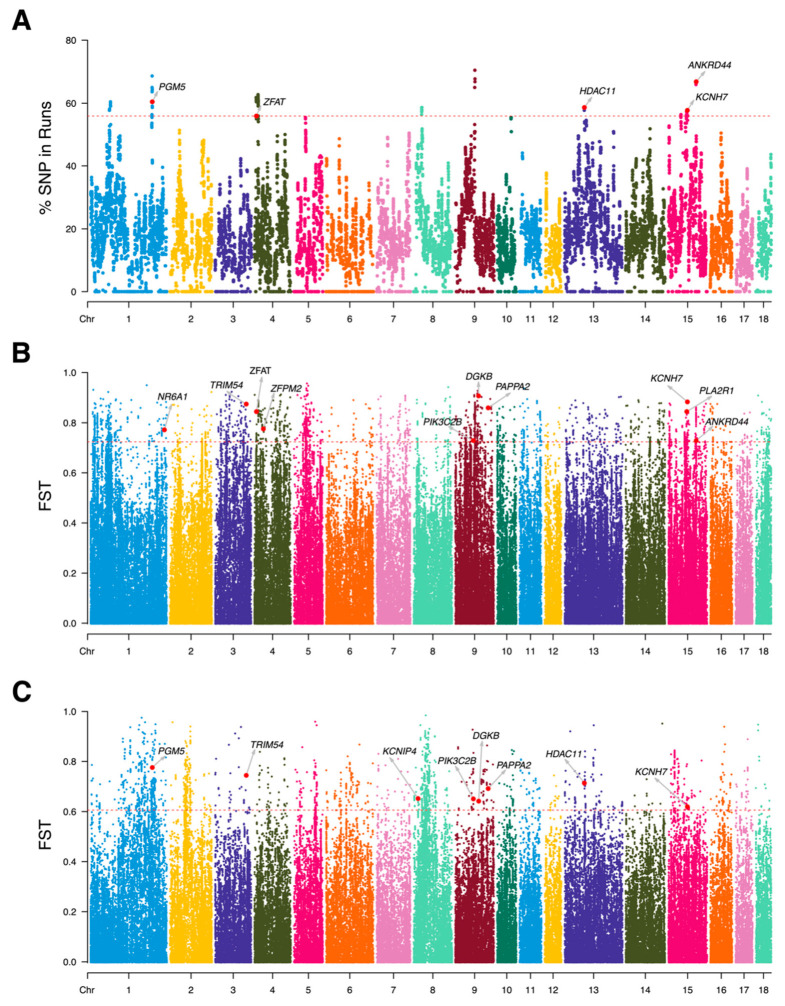
Manhattan plot collection. (**A**) Frequency of occurrences of each SNP within ROH regions in Licha black pig. The x-axis represents the locations of SNPs, and the y-axis represents the percentage of SNPs occurring within ROH regions. (**B**) The distribution of selection signatures between Licha black pigs and the other Chinese pigs was detected by *F_ST_*. The x-axis represents the locations of SNPs, and the y-axis represents the *F_ST_* values. (**C**) The distribution of selection signatures between Licha black pigs and commercial pigs was detected by *F_ST_*. The red line represents the top 1% threshold for statistical significance. Genes related to growth, muscle, and fat have been identified.

**Figure 4 biology-12-00500-f004:**
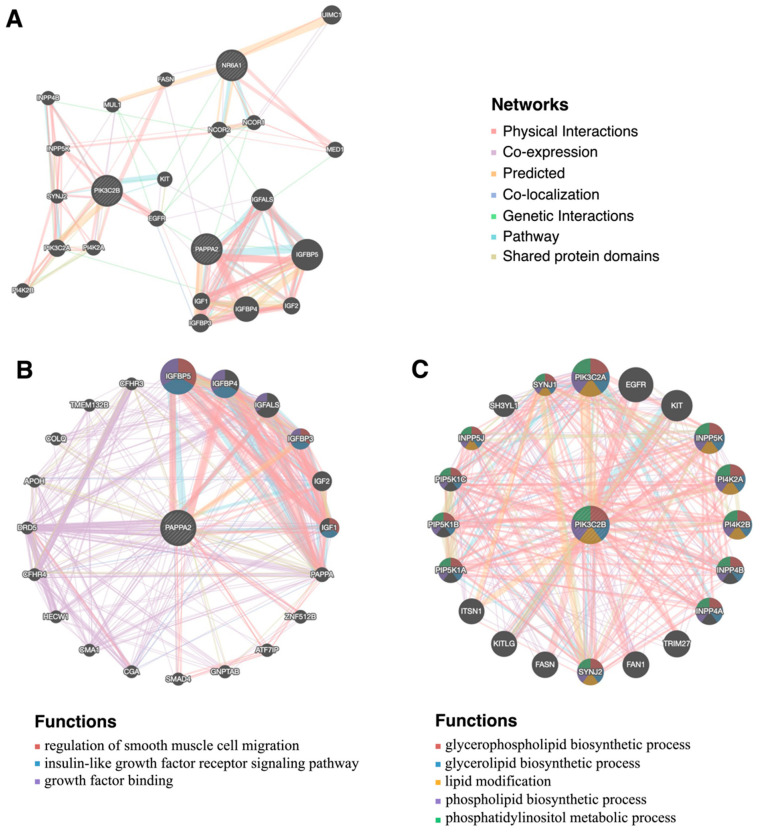
The PPI (protein–protein interaction) networks of *NR6A1*, *PAPPA2*, and *PIK3C2B* and its interacting protein partners. (**A**) The PPI networks of *NR6A1*, *PAPPA2*, and *PIK3C2B*. (**B**) The PPI networks of *PAPPA2* and its interacting protein partners and functions. (**C**) The PPI networks of *PIK3C2B* and its interacting protein partners and functions.

**Figure 5 biology-12-00500-f005:**
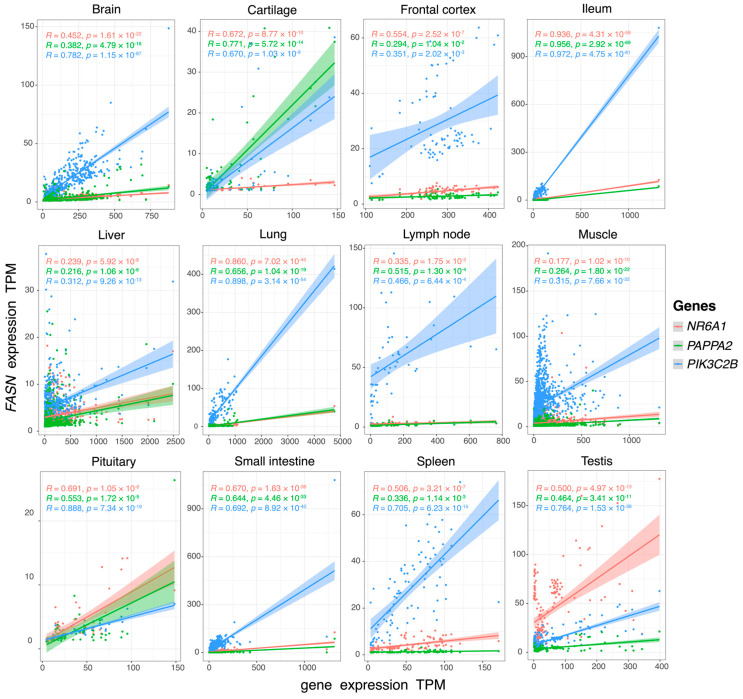
The correlation of gene expression between *NR6A1*, *PAPPA2*, *PIK3C2B*, and *FASN* across twelve pig tissues.

**Table 1 biology-12-00500-t001:** List of the genes selected by the Venn diagram that intersects with haplotype blocks on Licha black pig.

Genes	Chr ^a^	Gene Start (bp)	Gene End (bp)	Block Start (bp)	Block End (bp)	Block Length (Kb)	NSNPS ^b^
*KCNIP4*	8	15,211,916	16,346,170	15,865,991	16,065,990	200,000	102
	8	15,211,916	16,346,170	16,265,054	16,465,054	200,001	245
*LPP*	13	125,782,471	126,489,650	126,316,668	126,516,664	199,997	63
*IFT81*	14	31,368,687	31,657,305	31,422,148	31,622,148	200,001	119

^a^ Chr = Chromosome; ^b^ NSNPS = Number of SNPs.

**Table 2 biology-12-00500-t002:** Candidate genes in Licha black pigs that have been verified by previous studies, and the tissues with specific RNA expression in pigs.

Genes	Function	RNA Expression Tissue Specificity in Pigs [23]
(*F_ST_*-1∩*F_ST_*-2∩ROH)/Haplotype		
*KCNH7*	Glycerophospholipid pathway [28] Developmental delay syndrome [29]	brain
(*F_ST_*-1∩*F_ST_*-2)/(Haplotype∩ROH)		
*DGKB*	Early insulin secretion [30] Fat deposition [31] Short stature [32]	brain, salivary gland
*PAPPA2*	Longitudinal growth [33,34] Bone formation and dysplasia of the hip [35,36]	thyroid gland
*PIK3C2B*	Muscle-specific ablation [37] Lipid signaling [38]	/
*TRIM54*	Muscle signaling [39]	heart, mouth, skeletal muscle
(*F_ST_*-1∩Haplotype)/(*F_ST_*-2∩ROH)		
*LPP*	Smooth muscle expression [40]	smooth muscle
*IFT81*	Vertebrate developmental patterning [41] Skeletal dysplasias [42]	/
(*F_ST_*-1∩ROH)/(*F_ST_*-2∩Haplotype)		
*ANKRD44*	Skeletal muscle [43]	lymphoid tissue
*ZFAT*	Height [44]	/
(*F_ST_*-2∩Haplotype)/(*F_ST_*-1∩ROH)		
*KCNIP4*	Growth and development [45]	brain, retina, small intestine, smooth muscle
(*F_ST_*-2∩ROH)/(*F_ST_*-1∩Haplotype)		
*HDAC11*	Metabolic homeostasis and obesity [46,47] Control of adipose tissue [48] Skeletal muscle regeneration [49]	brain, testis
*PGM5*	Fetal growth restriction [50] Myofibril assembly and repair [51]	ductus deferens, smooth muscle, urinary bladder

FST-1, the *F_ST_* between Licha black pigs and other Chinese pigs; FST-2, the *F_ST_* between Licha black pigs and commercial pigs; ROH, the ROH islands within Licha black pigs; Haplotype, the haplotype blocks within Licha black pigs.

**Table 3 biology-12-00500-t003:** Base distribution, VEP variants type, and amino acid transformation of significant (*p*-value < 0.05) loci in *NR6A1*, *PAPPA2*, and *PIK3C2B*.

Gene	SNP	REF ^a^/ALT ^b^	Allele Numbers (REF/ALT)			Significance		Consequence	Impact	AA ^c^
			LI	Commercial	Chinese	LI-Commercial	LI-Chinese			
*NR6A1*	1:265347265	A/G	17/5	600/10	139/593	***	***	Missense variant	Moderate	L/P
*PAPPA2*	9:118365823	T/C	20/2	593/17	315/417		***	Missense variant	Moderate	S/P
	9:118365845	C/T	20/2	593/17	316/416		***	Missense variant	Moderate	A/V
	9:118365925	G/A	16/6	610/0	660/72	***	*	Missense variant	Moderate	D/N
	9:118365959	A/G	22/0	607/3	599/133		*	Missense variant	Moderate	K/R
	9:118366048	C/G	22/0	609/1	582/150		*	Missense variant	Moderate	P/A
	9:118549309	G/A	22/0	610/0	582/150		*	Missense variant	Moderate	R/K
	9:118500711	G/A	22/0	609/1	609/123		*	Missense variant	Moderate	A/T
*PIK3C2B*	9:65126980	G/C	17/5	485/125	692/40		**	Missense variant	Moderate	P/A
	9:65127090	G/A	17/5	482/128	694/38		**	Missense variant	Moderate	P/L
	9:65127444	G/A	17/5	485/125	692/40		**	Missense variant	Moderate	P/L

^a^ REF = Reference base; ^b^ ALT = Alternate base; ^c^ AA = Amino acids; * Significant difference *p* < 0.05; ** Significant difference *p* < 0.01; *** Significant difference *p* < 0.001.

## Data Availability

The whole genome re-sequencing datasets: 231 samples are available from the APLHADB (http://alphaindex.zju.edu.cn/ALPHADB/download.html, accessed on 25 March 2023), and 451 samples from previous studies are available from the NCBI (Bioproject number PRJCA001440, PRJEB1683, PRJEB37956, PRJEB38156, PRJEB39374, PRJEB9115, PRJEB9922, PRJNA144099, PRJNA186497, PRJNA213179, PRJNA238851, PRJNA239399, PRJNA255085, PRJNA260763, PRJNA305975, PRJNA309108, PRJNA322309, PRJNA343658, PRJNA369600, PRJNA378496, PRJNA398176, PRJNA41185, PRJNA438040, PRJNA487172, PRJNA488327, PRJNA488960, PRJNA493166, PRJNA506339, PRJNA524263, PRJNA531381, PRJNA550237, PRJNA553106, PRJNA622908, PRJNA626370, and PRJNA671763). The 50 K SNP chip datasets: 209 samples from previous studies are available from the FigShare Repository (https://figshare.com/articles/dataset/Analysis_of_genetic_diversity_and_population_structure_of_the_Licha_black_pig_population/15170826, accessed on 16 August 2021).

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
