# Peer review of "Integration of Selection Signatures and Protein Interactions Reveals NR6A1, PAPPA2, and PIK3C2B as the Promising Candidate Genes Underlying the Characteristics of Licha Black Pig"

_biology, 2023, doi:10.3390/biology12040500_

Round 1

Reviewer 1 Report

The paper deals with the integration of selection signatures and protein interactions and reveals NR6A1, PAPPA2 and PIK3C2B as the promising candidate genes underlying the characteristics of Licha Black Pig. The topic of the paper is current and interest and brings some new knowledge. However, I also have a few reservations about the manuscript, which are listed below.

Comments:

In Material and methods

-          in the section "Populations and Data" it is not clear whether the whole genome re-sequencing data were taken from other works or whether the research was carried out by the authors

-          if the research was carried out by the authors, there is no information on what biological material was used for DNA isolation, what method was used for DNA extraction, was the whole genome sequencing carried out at the authors' workplace or was it carried out by a commercial company .... - please add the following information to the Methods and Results

-          in the section “RNA expression specificity of candidate genes in human and pig tissues“- identify whether the RNA expression of selected candidate genes was tissue-specific in human is not relevant to this study

In Results

-          Table 2 – remove the last column from the table (RNA expression Tissue specificity in Human) - it is not relevant for this study, moreover, the authors do not discuss the information further

Author Response

Comments: The paper deals with the integration of selection signatures and protein interactions and reveals NR6A1, PAPPA2 and PIK3C2B as the promising candidate genes underlying the characteristics of Licha Black Pig. The topic of the paper is current and interest and brings some new knowledge. However, I also have a few reservations about the manuscript, which are listed below.

Response: Thank you for your advice regarding our manuscript. We are glad that you find the topic of our paper interesting and relevant. We appreciate your comments and have carefully considered the reservations you have listed below. We will address each of these issues in our revision and make the necessary changes to improve the quality of our manuscript. Once again, thank you for your valuable suggestion.

Comments 1: In Material and methods: in the section "Populations and Data" it is not clear whether the whole genome re-sequencing data were taken from other works or whether the research was carried out by the authors

- if the research was carried out by the authors, there is no information on what biological material was used for DNA isolation, what method was used for DNA extraction, was the whole genome sequencing carried out at the authors' workplace or was it carried out by a commercial company .... - please add the following information to the Methods and Results

Response 1: Thank you for your advice. The whole genome re-sequencing (WGS) data of 231 individuals generated by our laboratory, including 11 Licha black (LI) pigs, 23 commercial pigs, and 196 Chinese indigenous pigs. The WGS data of the remaining 451 individuals came from NCBI,including 282 commercial pigs and 169 Chinese indigenous pigs. The bioproject numbers are as followed: PRJCA001440, PRJEB1683, PRJEB37956, PRJEB38156, PRJEB39374, PRJEB9115, PRJEB9922, PRJNA144099, PRJNA186497, PRJNA213179, PRJNA238851, PRJNA239399, PRJNA255085, PRJNA260763, PRJNA305975, PRJNA309108, PRJNA322309, PRJNA343658, PRJNA369600, PRJNA378496, PRJNA398176, PRJNA41185, PRJNA438040, PRJNA487172, PRJNA488327, PRJNA488960, PRJNA493166, PRJNA506339, PRJNA524263, PRJNA531381, PRJNA550237, PRJNA553106, PRJNA622908, PRJNA626370, and PRJNA671763.

All experimental materials for lab WGS data were from ear tissue. Ear tissue was collected from the native pig conservation farm for high-throughput resequencing. We used the Qiagen DNeasy Tissue kit (Qiagen, Dusseldorf, Germany) to extract DNA samples from all pigs, and verified the integrity and purity of DNA by agarose gel electrophoresis and A260/280 ratio. Then, for sequencing and amplification, we processed the genomic DNA with the Covaris system end-repair, A-tailing, ligation of pair-ended adapter, and size-selection. Finally, we sequenced the amplified fragments on the HiSeqX platform using the scheme recommended by the manufacturer of Novogene (Beijing, China). The processing process refers to PHARP, briefly, we used BWA v0.7.17 software for mapping, SAMtools v1.10 software for sorting, and GATK v4.1.6 software for calling SNP and VariantFiltration. The criteria of VariantFiltration: "QD < 2.0, FS > 60.0, MQ <40.0, SOR > 3.0, MQRankSum < –12.5, ReadPosRankSum < –8.0".

We apologize for the vagueness of our initial description in the 'Materials and Methods' section. We have since included more specific information in lines 79-91 of the clean revised manuscript. Please refer to this section for further details.

Comments 2: In Material and methods: in the section “RNA expression specificity of candidate genes in human and pig tissues”- identify whether the RNA expression of selected candidate genes was tissue-specific in human is not relevant to this study

In Results: Table 2 – remove the last column from the table (RNA expression Tissue specificity in Human) - it is not relevant for this study, moreover, the authors do not discuss the information further

Response 2: Thank you for your suggestion. We agree that the section “RNA expression tissue specificity in human” is not relevant to our study, and we have therefore removed it from the current manuscript. Please refer to lines 156-158 and 293-294 of the clean revised manuscript. We appreciate your advice.

Reviewer 2 Report

The manuscript showed interesting results on signature of selection for Licha Black pigs. I have a concern about the haplotype block analysis since it is not clear about the assumptions behind it, and the steps to achieve it. Defining haplotypes are sensitive to the SNP numbers and distances, and the authors have prune the SNPs by LD  before haplotype based analyses?

Line 19: what are these characteristics,

Line 24: Why did the authors focus on FASN interaction?

Line 24: why in-depth, what is actual analysis?

Line 25: It is not clear what the authors mean by highly correlated.

Line 32-33: The authors might add more information about the origin of the LL pigs and how selection takes place for this breed. What is the current status of the breed etc.

Line 35: Lower and larger compared to which breeds?
Line 41: How many lineages did this breed have, and how many disappear?

Line 44: LI pigs or LI pig, I think the plural form should be correct, so keep the consistency and make the necessary changes.

Line 47: ROH based on which types of genomic data, SNP chip or WGS?

Line 47-48: give references and which methods of identification of candidate genes here.

Line 63: remove hot, what did the authors mean here,

How many SNPs are from whole genome resequencing?

What is accurate of imputation with Beagle, the authors should remove SNPs with low imputation accuracy before performing other QC controls

How is the QC for the HW test?

Line 88: What is the theory for removing the SNPs with this LD< the authors remove 90% of SNPs after QC, it did not make sense.

Line 107-111: It is not clear the motivation for Haplotype analyses here, especially since the authors have removed the SNPs with LD.

Line 123: I do not think the authors need to use the Human database here, this database is derived from cancer research. The data from a pig is enough.

The results and discussion are fine,

Line 367: Change AA to amino acid

Line 399 to 402: I do not see the usefulness of this conclusion. What/why did the authors want readers to know about it?

Author Response

Comments 1: The manuscript showed interesting results on signature of selection for Licha Black pigs. I have a concern about the haplotype block analysis since it is not clear about the assumptions behind it, and the steps to achieve it. Defining haplotypes are sensitive to the SNP numbers and distances, and the authors have prune the SNPs by LD before haplotype based analyses?

Response 1: Thank you for your advice on our manuscript, and we appreciate your comments and concerns about our haplotype block analysis. Based on your suggestion, we have conducted haplotype analysis again using SNPs without LD pruning. As a result, we have updated the "Materials and Methods" section accordingly. Please refer to lines 107-108, 130-144 of the clean revised manuscript.

The results of the haplotype analysis have also been updated. In the LI population, there are now 14,482 haplotype blocks (previously 3,064) containing 264,823 SNPs (previously 10,663), which cover a genome range of 707157.385 Kb (previously 138854.343 Kb). Please see lines 274-270 for a more detailed description of the haplotype block analysis results. Additionally, we have identified more blocks and genes such as LPP and IFT81. Please refer to lines 231-236, 255-258 and 291-294 (Table 1-2) of the clean revised manuscript for further details.

Comments 2: Line 19: what are these characteristics,

Response 2: Thank you for your comment regarding line 19. The Licha black (LI) pig is a Chinese indigenous pig breed known for its distinct characteristics of larger body length and appropriate fat deposition. We mentioned these characteristics of Licha black pig in line 15-16.

Comments 3: Line 24: Why did the authors focus on FASN interaction?

Response 3: Thank you for your advice. Our analysis revealed that NR6A1, PAPPA2, and PIK3C2B are associated with Fatty Acid Synthase (FASN) through Protoin-Protein Interaction (PPI) networks. These associations involve physical interaction and co-expression, which suggests that the expression of FASN may indirectly reflect the expression of genes that interact with it. Our study of FASN also relates to our desire to further investigate the genetic characteristics of Licha black pigs.

Comments 4: Line 24: why in-depth, what is actual analysis?

Response 4: Thank you for your advice. To further confirm the association between NR6A1, PAPPA2, PIK3C2B, and FASN are indeed related, as well as identify the possible tissues of gene interaction, we utilized RNA expression data from FarmGTEx (https://www.farmgtex.org/) to calculate the correlation coefficient of TPM gene expression between three genes (NR6A1, PAPPA2, and PIK3C2B) and FASN across pig tissues. We have modified this part in the current manuscript, please check the line 24-25 in the clean revised manuscript.

Comments 5: Line 25: It is not clear what the authors mean by highly correlated.

Response 5: Thank you for your advice. The RNA expression data from FarmGTEx provided evidence that the RNA expression levels of NR6A1, PAPPA2, PIK3C2B, and FASN were correlated in intestinal tissues. We have modified this part in the current manuscript, please check the line 24-25 in the clean revised manuscript.

Comments 6: Line 32-33: The authors might add more information about the origin of the LL pigs and how selection takes place for this breed. What is the current status of the breed etc.

Response 6: Thank you for your advice. The Licha black (LI) pig is an exceptional local breed of pig found in Shandong Province, with strong ancestral ties to other indigenous Chinese pig breeds, particularly those originating from Shandong Province. The book "Livestock Breeds in Shandong Province" indicates that the Licha black pig has been domestically bred for over two millennia. However, with the introduction of commercial pig breeds such as Landrace and Yorkshire into China during the 1940s and 1970s, the population of LI pigs has significantly decreased, with some lineages being entirely lost. As a result, there is a growing interest in studying the genetic traits of LI pigs to preserve this valuable breed. We have modified this part in the current manuscript, please check the line 32-42 in the clean revised manuscript.

Comments 7: Line 35: Lower and larger compared to which breeds?

Response 7: Thank you for your advice. Studies have demonstrated that LI pigs possess lower fat deposition and longer body length than other indigenous pig breeds in China. This unique combination of traits makes LI pigs particularly valuable and motivates us to conduct a thorough comparison of their characteristics with other Chinese indigenous pig breeds. Our goal is to gain a comprehensive understanding of the distinctive features of LI pigs and their potential uses. We have revised this section in the current manuscript. Please review the updated lines 37-38 in the clean revised manuscript.

Comments 8: Line 41: How many lineages did this breed have, and how many disappear?

Response 8: Thank you for your advice. Based on the research conducted by Wang et al. (DOI: 10.3390/ani12081045), it has been established that LI pigs have nine familial lineages. Our results are basically consistent with the report, but we have identified one additional lineage, which may be the larger group we studied. Unfortunately, we do not have information about the number of individuals that may have disappeared.

Comments 9: Line 44: LI pigs or LI pig, I think the plural form should be correct, so keep the consistency and make the necessary changes.

Response 9: Thank you for your advice. After careful consideration, we will unify the form in the current manuscript and use the plural form of "LI pigs".

Comments 10: Line 47: ROH based on which types of genomic data, SNP chip or WGS?

Response 10: Thank you for your advice. ROH based on WGS and imputed SNP chip data. The following is how we process and use these data:

Using WGS data as a panel, we imputed chip data of 209 LI pigs with Beagle v5.4 software and obtained 53,789,547 SNPs. The imputed chip data was filtered with BCFtools v1.10.2 for SNPs with DR2 less than 0.3 and merged with the WGS data. Next, PLINK v1.9 software was used for quality control, and unqualified individuals and SNP sites were eliminated. The criteria were as follows: (1) Genotype detection rate and individual detection rate were greater than 95%; (2) The loci with the minimum allele frequency (MAF) less than 0.05 were filtered out; (3) Remove SNPs on sex chromosomes. The combined data contained 277,882 high-confidence SNPs and 891 individuals, which was used for ROH analysis.

Comments 11: Line 47-48: give references and which methods of identification of candidate genes here.

Response 11: Thank you for your advice. We are sorry for the lack of references and methods. Now we have added more specific information here. Please see line 49-52 of the clean revised manuscript.

Comments 12: Line 63: remove hot, what did the authors mean here,

Response 12: Thank you for your advice. We originally hoped to use the word "hot" to express our affirmation of the current analysis results. We are sorry to confuse you. Now, we have revised this sentence in the current manuscript. Please see line 62-64 of the clean revised manuscript.

Comments 13: How many SNPs are from whole genome resequencing?

Response 13: Thank you for your advice. The lab WGS data retained a total of 53,789,547 SNPs were detected, including 231 individuals. The downloaded WGS data of 451 individuals was processed by GATK v4.1.6, and 53,789,547 SNPs were obtained, the same as the lab WGS data.

When we use WGS data for candidate genes’ screening, we filtered WGS data with PLINK v1.9 (command: PLINK --geno 0.1 --mind 0.1 --maf 0.05), and 28,490,962 SNPs were retained in total.

We mentioned complex data usage in our manuscript, and to aid your understanding, we have revised the manuscript (line 93-111 in the clean revised manuscript) and included a graphical abstract.

Comments 14: What is accurate of imputation with Beagle, the authors should remove SNPs with low imputation accuracy before performing other QC controls

Response 14: Thank you for your advice. We apologize for not including the information about filtering SNPs with DR2 less than 0.3 in our original manuscript. We have now added detailed records about this step in the new version of the manuscript. Please refer to line 100-101 in the clean revised manuscript.

Comments 15: How is the QC for the HW test?

Response 15: Thank you for your advice. We did not perform Hardy-Weinberg equilibrium (HWE) filtering in our study. Due to the limited population size of the indigenous pig population, HWE filtering may introduce bias. We also referred to previous studies, such as Chen et al. and Wang et al., who did not apply HWE filtering in their detection of selection signatures in Chinese indigenous pigs. (DOI: 10.3390/genes13122310 and DOI: 10.1038/s41598-022-14686-w, respectively).

Comments 16: Line 88: What is the theory for removing the SNPs with this LD< the authors remove 90% of SNPs after QC, it did not make sense.

Response 16: Thanks for your advice. To minimize ascertainment bias, we used 29,934 pruned SNPs with LD<0.4 for ADMIXTURE analysis. We referred to studies like Wang et al. who used 25,839 LD-filtered SNPs for the same purpose (DOI: 10.1111/eva.13124).

Comments 17: Line 107-111: It is not clear the motivation for Haplotype analyses here, especially since the authors have removed the SNPs with LD.

Response 17: Haplotype blocks, which are chromosome fragments with high linkage and low recombination rates, can reflect differences in genome structure. They can be passed down to the next generation as a whole and are affected by recombination, selection, and population hybridization. The distribution of haplotype blocks in the genome can reflect the genetic structure and variation of a population.

In our study, haplotype analysis served two important purposes: (1) identifying breed-specific genes in LI pigs, such as LPP and IFT81, and (2) focusing on shared genes identified by multiple methods (ROH, haplotype, and FST) to reduce the false positive rate of gene screening.

Based on your suggestion, we re-analyzed the haplotype analysis using non-LD-filtered data, and this led to some changes in our results, which we have detailed in response 1.

Comments 18: Line 123: I do not think the authors need to use the Human database here, this database is derived from cancer research. The data from a pig is enough.

Response 18: Thank you for your suggestion. We agree that the section “RNA expression tissue specificity in human” is not relevant to our study, and we have therefore removed it from the current manuscript. Please refer to lines 156-158 and 293-294 of the clean revised manuscript. We appreciate your advice.

Comments 19: Line 367: Change AA to amino acid

Response 19: Thank you for your comment regarding line 367. We agree that using "amino acid" instead of "AA" would improve the clarity of our manuscript. We have made the necessary changes as per your suggestion and appreciate your advice.

Comments 20: Line 399 to 402: I do not see the usefulness of this conclusion. What/why did the authors want readers to know about it?

Response 20: Thank you for your advice regarding line 399 to 402. We have deleted this sentence in our current manuscript.

Reviewer 3 Report

This is an interesting manuscript aimed to study candidate genes associated with growth and fat deposition traits in Licha black pigs. However, molecular methods involved in the study should be described in more detail to make clearer how the results were obtained. I suggest to consider next minor comments related to Materials and Methods section:

1)      Data.

-          Define the growth and fatness traits that were included in the study and describe how were they measured.

2)      Whole genome re-sequencing data.

-          Please describe briefly the processes involved in re-sequencing (i.e., DNA isolation, library construction, sequence alignment, etc.).

-          Explain statistical procedures used to identify SNPs associated with growth and fatness traits.

3)      50 k SNP chip data.

-          Please describe briefly DNA extraction and genotyping processes.

-          Indicate what was the test used to correct for multiple SNP testing, as well as the threshold p-value for significant SNPs after correction.

-          Explain statistical procedures used to identify SNPs associated with growth and fatness traits.

4)      RNA expression.

-          Please describe briefly the processes of RNA isolation, RNA sequencing and gene expression analyses.

Author Response

Comments: This is an interesting manuscript aimed to study candidate genes associated with growth and fat deposition traits in Licha black pigs. However, molecular methods involved in the study should be described in more detail to make clearer how the results were obtained. I suggest to consider next minor comments related to Materials and Methods section.

Response: Thank you for your affirmation of my current research. I am very proud to bring you inspiration. Of course, I also appreciate your comments. My response to the comments is as follows.

Comments 1: Data.

-          Define the growth and fatness traits that were included in the study and describe how were they measured.

Response 1: Thank you for your advice. In our study, our aim was to understand the characteristics of the LI breed. We utilized ROH, haplotype and FST selection signals to identify germplasm-specific SNPs and annotate genes in the LI breed, compared to other Chinese pig breeds and commercial pig breeds. By combining literature reports and the Pig RNA Atlas (http://rnaatlas.org/) (DOI: 10.1186/s12915-022-01229-y), we coincidentally discovered that some of these genes are related to growth and fatness traits.

It is important to note that our manuscript did not include a direct measurement of the LI breed's growth and fatness traits. However, we learned from sources such as "Livestock Breeds in Shandong Province" and "Genetic evidence for the introgression of Western NR6A1 haplotype into Licha breed associated with increased vertebral number" by Yang et al. (DOI: 10.1111/j.1365-2052.2008.01820.x) that the LI breed has a larger body length (21.5 vertebrae on average) and appropriate fat deposition than other pigs in China.

Comments 2: Whole genome re-sequencing data.

-          Please describe briefly the processes involved in re-sequencing (i.e., DNA isolation, library construction, sequence alignment, etc.).

-          Explain statistical procedures used to identify SNPs associated with growth and fatness traits.

Response 2: Thank you for your advice.

All experimental materials for lab WGS data were from ear tissue. Ear tissue was collected from the native pig conservation farm for high-throughput resequencing. We used the Qiagen DNeasy Tissue kit (Qiagen, Dusseldorf, Germany) to extract DNA samples from all pigs, and verified the integrity and purity of DNA by agarose gel electrophoresis and A260/280 ratio. Then, for sequencing and amplification, we processed the genomic DNA with the Covaris system end-repair, A-tailing, ligation of pair-ended adapter, and size-selection. Finally, we sequenced the amplified fragments on the HiSeqX platform using the scheme recommended by the manufacturer of Novogene (Beijing, China). The processing process refers to PHARP, briefly, we used BWA v0.7.17 software for mapping, SAMtools v1.10 software for sorting, and GATK v4.1.6 software for calling SNP and VariantFiltration. The criteria of VariantFiltration: "QD < 2.0, FS > 60.0, MQ <40.0, SOR > 3.0, MQRankSum < –12.5, ReadPosRankSum < –8.0". We are sorry what we wrote in "Material and Methods" is vague. Now we have added more specific information in "Materials and Methods". Please see line 79-92 of the clean revised manuscript.

In our current research, our main purpose is to understand the characteristics of the LI breed. Due to the absence of phenotype, we did not conduct an association analysis in the current study. However, we performed a signature of selection analysis for the LI breed using methods such as ROH, haplotype and FST. Further details about our methodology have been briefly described in response 1.

Comments 3: 50 k SNP chip data.

-          Please describe briefly DNA extraction and genotyping processes.

-          Indicate what was the test used to correct for multiple SNP testing, as well as the threshold p-value for significant SNPs after correction.

-         Explain statistical procedures used to identify SNPs associated with growth and fatness traits.

Response 3: Thank you for your advice. The 50 K SNP chip data was downloaded from Figshare, and therefore, DNA extraction and genotyping were not conducted. However, we have processed lab WGS data in detail, and the processing method has been mentioned in detail in response 2. For 50k chip data, we have done the following: Using WGS data as a panel, we imputed chip data of 209 LI pigs with Beagle v5.4 software and obtained 53,789,547 SNPs. The imputed chip data was filtered with BCFtools v1.10.2 for SNPs with DR2 less than 0.3 and merged with the WGS data. Next, PLINK v1.9 software was used for quality control, and unqualified individuals and SNP sites were eliminated. The criteria were as follows: (1) Genotype detection rate and individual detection rate were greater than 95%; (2) The loci with the minimum allele frequency (MAF) less than 0.05 were filtered out; (3) Remove SNPs on sex chromosomes. The combined data contained 277,882 high-confidence SNPs and 891 individuals, which was used for ROH, haplotype and FST selection signatures. For combined data, discard those whose LD (linkage imbalance) was greater than 0.4 in these populations (command: PLINK --indep-pair 50 10 0.4), there were 29,934 SNPs left in total. The LD-filtered combined data was used for population structure and genetic diversity. We apologize for any confusion caused by the lack of clarity in the manuscript.

In our new study, our aim is to investigate the characteristics of the LI breed and we did no correct for multiple SNP testing and we are unsure about the threshold p-value for significant SNPs after correction. However, some methods we used in our study, such as ROH, haplotype, and FST, required a threshold and p-value for analysis. Specifically, we set the following thresholds: (1) we selected the SNPs with the top 1% frequency in all individual ROH regions, (2) we selected the top 1% haplotype blocks with the longest length, (3) we selected the SNPs with the top 1% FST value. Additionally, we used Fisher's exact test to determine if the distribution of base corresponding to each locus allele of LI breed and other breeds were significantly different, and reported results with a p-value < 0.05. Finally, we calculated the correlation of gene expression level between NR6A1, PAPPA2, PIK3C2B, and FASN in different pig tissues and reserved results with a p-value < 0.05.

In our current research, our primary objective is to investigate the characteristics of the LI breed. Our current study does not involve statistical procedures for identifying SNPs associated with growth and fatness characteristics. Our research methods have been described in more detail in response 1.

Comments 4: RNA expression.

-          Please describe briefly the processes of RNA isolation, RNA sequencing and gene expression analyses.

Response 4: Thank you for your advice. The RNA expression data used in our manuscript was obtained from The Farm animal Genotype-Tissue Expression (FarmGTEx) database (https://www.farmgtex.org/) (DOI: 10.1101/2022.11.11.516073). We have cited this data source in the manuscript. Please see line 180-184 of clean revised manuscript.

Round 2

Reviewer 2 Report

The authors have addressed my comments. Below are some minor suggestions:

Line 25: I suggest changing the word proved since the correlation of expression does not mean the causal relationship. In addition, it is not clear what the authors want to prove here. Did the authors check if the correlation is significant or not?
Line 84: Write exact the name of the chip,

Line 110: Which methods did the authors use for variant calling, it seems that the number of called variants is very high (56 Mil)?

Line 157: The authors might write detail about what is the DR2

Line 168: The authors might give the support references for choosing a threshold of 0.4.

Table 1: Add the footnote to explain what are Chr and NSNPS

Table 2: Remove humans in the title

Line 305-312: Add the comma for the numbers such as  707,157.385

Line 636-648; The authors might use the abbreviation for the Licha breed to keep it consistent.

Author Response

Comments 1: Line 25: I suggest changing the word proved since the correlation of expression does not mean the causal relationship. In addition, it is not clear what the authors want to prove here. Did the authors check if the correlation is significant or not?

Response 1: Thank you for your advice. We agree that the use of "indicated" instead of " proved" would be more appropriate, given that correlation does not necessarily imply causation. Our aim was to establish a significant correlation between the RNA expression levels of three genes (NR6A1, PAPPA2, and PIK3C2B) and FASN. We did, in fact, perform statistical analysis to check the significance of this correlation, and our results confirm its statistical significance. Our study demonstrates a positive correlation among three genes (NR6A1, PAPPA2, and PIK3C2B) and FASN in various pig tissues. Notably, the correlation was high in the ileum (ranging from 0.936 to 0.972), and moderate in the small intestine (ranging from 0.644 to 0.692), cartilage (ranging from 0.670 to 0.771) and lung (ranging from 0.656 to 0.898).

Comments 2: Line 84: Write exact the name of the chip,

Response 2: Thank you for your advice. The exact name of the chip is “Zhongxin-I” Porcine 50k SNP Chip (Beijing Compass Agritechnology Co., Ltd., Beijing, China). We have written this information in the manuscript. Please see line 69-70 of clean revised manuscript.

Comments 3: Line 110: Which methods did the authors use for variant calling; it seems that the number of called variants is very high (56 Mil)?

Response 3: Thank you for your advice. We utilized PHARP (http://alphaindex.zju.edu.cn/PHARP/index.php/) (DOI: 10.1038/s41598-022-15851-x) as a panel, which comprises 2,048 individuals and allows for the detection of a large number of SNPs. We subsequently conducted a follow-up screening of the LI population, resulting in the identification of 28,490,962 SNPs for further analysis. Please check line 82-96 of clean revised manuscript.

The specific steps for data processing and variant discovery were as follows: SRA Toolkit (https://github.com/ncbi/sra-tools) was used to download (prefetch) WGS data and convert (fasterq-dump) them from SRA to FASTQ format. Quality control, read filtering and base correction for the raw FASTQ data were performed using fastp with default parameters. The high-quality reads were mapped to the latest version of the pig reference genome (Sscrofa11.1) using BWA v0.7.17 with the MEM function and parameters for paired-end data. SAM files were converted to BAM files, and library data from individual and multiple experiments were merged into one dataset using samtools v1.10. Duplicated reads were removed with sambamba v0.7.1. Coverage and depth were individually calculated using Mosdepth v0.2.9, and finally, GATK v4.1.6 HaplotypeCaller was applied to each sample to generate an intermediate GVCF, which was then used in GenotypeGVCFs for joint genotyping across all samples.

Comments 4: Line 157: The authors might write detail about what is the DR2

Response 4: Thank you for your advice. DR2 (Dosage R-Squared) refers to the estimated squared correlation between estimated REF dose [P(RA) + 2*P(RR)] and true REF dose. We have provided this information in the manuscript and it can be found on line 105 of clean revised manuscript.

Comments 5: Line 168: The authors might give the support references for choosing a threshold of 0.4.

Response 5: Thank you for your advice. As you correctly noted, Badke et al. (DOI: 10.1186/1471-2164-13-24) have shown that high average LD (r2 > 0.4) between adjacent SNP is an important precursor for the implementation of marker assisted selection within a livestock species. In our study, we used this threshold of 0.4 for LD filtering, as we believe it allowed us to obtain independent sites and perform more efficient analysis. We appreciate your attention to the literature and hope that this explanation clarifies our methodology.

Comments 6: Table 1: Add the footnote to explain what are Chr and NSNPS

Response 6: Thank you for your advice. Just to clarify, "Chr" refers to "chromosome," and "NSNPS" stands for "number of SNPs." We have included this information in the manuscript, and it can be found on line 294 of the revised version. We appreciate your attention to detail and hope that this explanation helps to clarify any confusion.

Comments 7: Table 2: Remove humans in the title

Response 7: Thank you for your advice. We appreciate your suggestion to remove the word "humans" from the title of Table 2. We have made this change in the revised manuscript, and you can find it on line 296. Thank you for your attention to detail.

Comments 8: Line 305-312: Add the comma for the numbers such as 707,157.385

Response 8: Thank you for your advice. We have made this change in the revised manuscript and you can find it on lines 234-242.

Comments 9: Line 636-648; The authors might use the abbreviation for the Licha breed to keep it consistent.

Response 9: Thank you for your advice. We appreciate your suggestion to use an abbreviation for the Licha breed to maintain consistency in our manuscript. We have made the necessary changes throughout the manuscript. Thank you for your attention to detail.
